# Mother–Infant Co-Sleeping and Maternally Reported Infant Breathing Distress in the UK Millennium Cohort

**DOI:** 10.3390/ijerph17092985

**Published:** 2020-04-25

**Authors:** David Waynforth

**Affiliations:** School of Medicine, Bond University, Gold Coast 4229, Australia; dwaynfor@bond.edu.au

**Keywords:** Neonatal health, infant sleep, safe sleeping, machine learning, cohort studies, SIDS, SUDI

## Abstract

Mother–infant co-sleeping or bed sharing is discouraged by health organisations due to evidence that it is associated with unexplained sudden infant death. On the other hand, there is evidence that it should theoretically be beneficial for infants. One line of this evidence concerns breathing regulation, which at night is influenced by the rocking movement of the mother’s chest as she breathes. Here, the hypothesis that mother–infant co-sleeping will be associated with a lower probability of infant breathing distress is tested in the UK Millennium Cohort Study (n = 18,552 infants). Maternal, infant, family, and socio-economic covariates were included in logistic regression analysis, and in a machine learning algorithm (Random Forest) to make full use of the number of variables available in the birth cohort study data. Results from logistic regression analysis showed that co-sleeping was associated with a reduced risk of breathing difficulties (OR = 0.69, *p* = 0.027). The Random Forest algorithm placed high importance on socio-economic aspects of infant environment, and indicated that a number of maternal, child, and environmental variables predicted breathing distress. Co-sleeping by itself was not high in the Random Forest variable importance ranking. Together, the results suggest that co-sleeping may be associated with a modest reduction in risk of infant breathing difficulties.

## 1. Introduction

Mother–infant bed co-sleeping is widely practiced around the world, and is almost certainly considered unremarkable and uncontroversial in the vast majority of cultures [1,2]. However, in North America and Western Europe it is less universally practised, and became medicalised due to studies showing evidence of a link with sudden unexpected death in infancy (SUDI) [3,4]. Despite the medicalization of co-sleeping in North American and Western European societies, it remains a routine sleeping arrangement in half or more households [5,6]. Health organisations and health systems have responded with guidelines ranging from discouraging co-sleeping entirely to attempting to ensure that parents who choose to co-sleep with their infant do so in the safest way possible given research evidence of specific sleeping practices and SUDI risk. The United States Centers for Disease Control (CDC) and some UK National Health Service trusts have published guidelines suggesting that babies should sleep in a crib or cot rather than in bed with their parent(s) [7,8], and highlighting risks of infant suffocation and strangulation in bed [6]. In some countries, boxes are distributed to new mothers as a method of separating mothers and infants in the parental bed [9]. A number of health organizations have not gone as far as discouraging bed co-sleeping, and instead attempt to support parents who wish to co-sleep by publishing safety guidelines [10,11]. The purpose of the present research was to explore a counter-argument to the North American and Western European practice of discouraging mother–infant co-sleeping or highlighting its risks. The hypothesis here is that for the majority of infants, mother–infant co-sleeping is likely to provide health benefits. If this is the case, SUDI guidance would need to be balanced with safe co-sleeping guidance for parents to promote infant health.

The global distribution of cultures in which the norm is for infants to sleep alone suggests that mother–infant physical separation at night is historically recent, and is part of a model of infant care which emphasises early independence currently found in many Western European societies and nations colonised by Western European people in the last few hundred years [12]. Mother–infant separation at night is uncommon in other cultures [13]. Given the historical and global ubiquity of parent-infant co-sleeping, it would be surprising for it to be associated with negative health outcomes for infants. From an evolutionary perspective, natural selection would very likely have already led to behavioural change including a desire not to sleep with infants if there were any significant risks associated with it. Nevertheless, evidence from case-control studies and from studies of changes in the patterns of SIDS before and after public health ‘safe sleeping’ campaigns suggest a link between co-sleeping and SUDI [14,15,16]. Whether the relationship is causal is unclear, as co-sleeping, poverty, sofa or couch sleeping, and maternal alcohol consumption are often correlated in SUDI cases [14]. In addition, sample sizes of SUDI cases included in statistical analyses can be quite small due to the rarity of SUDI, which occurs in less than one in a thousand births [3,14].

While the bulk of research and parenting guidelines have focussed on SUDI risk, there are a number of ways in which co-sleeping may be beneficial for infants. Studies of infant stress responses have shown that babies who sleep alone have a greater cortisol-mediated stress response both at home and in experimental conditions than infants who co-sleep [17,18,19]. For example, in Ainsworth’s Strange Situation test, which tests the reaction of children to being separated from their mother, children who co-slept with their mother showed less distress and had lower cortisol levels than solitary sleepers [18].

Co-sleeping appears to have a range of effects on infants in addition to being a predictor of the stress response. Co-sleeping and close maternal-infant physical contact mediate physiological processes as well as socio-emotional development [20]. Trevathan and McKenna’s review of these processes highlighted a number of studies which focus on infant breathing: infants respond to human breathing sounds by breathing, and infant regulation of breathing is influenced by the rocking movement of their parent’s chest while breathing. Breathing is regulated in a coordinated pattern with suckling and swallowing during breastfeeding [20,21], and if breastfeeding occurs at night in conjunction with co-sleeping, this breathing regulatory pattern will be occurring. In addition, if a baby is close to its mother’s exhaled air, the change in partial pressure of carbon dioxide should theoretically produce a breathing response.

Given evidence of links between close maternal contact and the physiology of infant breathing regulation outlined above, the hypothesis tested in the present research was that co-sleeping infants will be less likely to experience breathing difficulties. Because breathing regulation is likely to be most critical very early in infancy, when regulation is still developing [22], the focus of the analysis was early infancy (the first week from birth).

## 2. Materials and Methods

### 2.1. Sample

The hypothesis was tested using data drawn from the UK Millennium Cohort Study (henceforth MCS). The MCS data are freely available via the UK Data Service. The MCS was designed as a birth-to-death longitudinal cohort study, and represented a continuation of the UK’s series of national cohort studies begun in 1946. The MCS consisted of 18,552 infants born from September 2000 to August 2001. The first sweep of interviews with cohort members’ mothers (or in a few cases other main care providers) took place when the infants were around 9 months old. The interviews included questions on a wide variety of topics, including health, education, social, family and economic status of the cohort members’ households. Detailed information on the sampling and scope of the UK Millennium Cohort study is available in a published cohort profile [23].

The MCS has two key advantages for testing the hypothesis that co-sleeping is linked with fewer breathing difficulties. First, it is likely to provide enough cases for the statistical analysis of infants for whom breathing difficulties were reported. Second, it is possible to include a wide range of covariates so that co-sleeping can be included in statistical models which explain a reasonable amount of the variation in the occurrence of breathing difficulties. These advantages come at a cost to detailed information on specific co-sleeping behaviours and patterns which could be gained from collecting new data, and to detail about the specifics of the breathing difficulties that occurred.

### 2.2. Variables and Statistical Analysis

The outcome variable was maternal reports of infant breathing difficulties up to a week from birth. Mothers, or in a small number of cases the main infant care provider if it was not the biological mother, were asked to provide information to an interviewer on all health problems that occurred in the first week from birth. Two questions on breathing difficulties were included: breathing delay at birth, and breathing difficulties in the first week. The latter was used as a binary outcome variable. The main predictor variable was whether or not the infant co-slept with a parent in bed as their normal sleeping arrangement. The mother’s first response to the interview question was used, as the aim was to capture the main sleeping arrangement, rather than a sleeping arrangement that was not the most typical for the infant on any given night. Table 1 contains a summary of all variables included in the analyses including why they were selected.

Two approaches to data analysis were taken. First, logistic regression, including co-sleeping status with covariates which have demonstrated importance for infant health, and which may themselves be correlated with sleeping arrangements (see Table 1). The regression modelling priority was to minimize model complexity but include important predictors of infant health from the published literature. All main effects of covariates were forced into the model. One interaction effect was forced into the model: the interaction between co-sleeping and breastfeeding. This was done because co-sleeping and breastfeeding together place the infant in a sleeping position which should bring the hypothesised mechanisms into play, such as the infant being in contact to feel the rocking motion of the mother’s breathing [35]. To minimise unwanted model complexity, the other two-way interactions between co-sleeping and covariates were dropped from the model using a stepwise approach if they did not reach the statistical significance threshold of *p* < 0.05. Squared terms were included for predictors with significant quadratic terms in initial exploratory analyses. Interaction effects were residual centred to avoid collinearity with the variables that they were constructed from.

The second approach was to make use of MCS variables which do not have well-established links with infant health, but which nevertheless are plausible predictors of infant breathing distress. A machine learning classification tree algorithm was run (Random Forest, henceforth RF) [36]. The priority with the machine learning approach was to maximise classification accuracy in predicting breathing distress including a range of variables representing socioeconomic position, social, family, and circumstances surrounding pregnancy and birth. The RF algorithm approach can accommodate a large number of variables without resulting in model over-specification or overfitting, and hence is ideal when a large number of potentially relevant variables are present [36]. In addition, two-way interaction effects are automatically included: if a variable is more important in the context of other variables included in the model, it will receive a higher variable importance score. The importance of co-sleeping was determined by outputting variable importance statistics for the model. Variable importance was established using the default method in the *rforest* Stata program, which uses the change in predictive accuracy when each variable is removed. A disadvantage of using the RF classification tree approach is that it does not yield statistics familiar to most readers of research in the social and health sciences: there are no *p*-values or 95% confidence intervals to report. However, both logistic regression and RF report correct and incorrect classification of cases. In RF, this is a default part of the output. In logistic regression, a classification table can be produced after running the analysis. The classification table is used here to compare the likely predictive accuracy of the two analysis approaches. All analyses were carried out using Stata, version 16.

## 3. Results

### 3.1. Descriptive Statistics

Descriptive statistics for the study variables are reported in Table 2.

### 3.2. Logistic Regression Results

A correlation matrix was produced prior to carrying out the logistic regression analysis to examine the risk of multicollinearity in the model. Father presence in the household, income adjusted using McClement’s equivalency, maternal age and maternal education were correlated at r = 0.25 or higher: older mothers had completed more education (r = −0.25) and had higher income scores (r = 0.35). Father absence was correlated with a lower income score (r = 0.32). These close associations are likely to increase the variance for each of these variables in the regression model and lead to underestimation of their importance. There were no close correlations between co-sleeping and the other covariates, hence the regression estimate for co-sleeping will be unaffected. Appendix A contains the correlation matrix for all predictors. The RF model is likely to provide better estimates for the socio-economic variables and maternal age, as collinearity is not a problem in RF algorithms.

Table 3 displays the results of the final logistic regression model. None of the co-sleeping interactions with covariates reached statistical significance, and hence these are not shown, with the exception of the interaction between co-sleeping and breastfeeding, which formed part of the hypothesis. Co-sleeping was statistically significantly associated with a reduced risk of breathing difficulties, but its interaction with breastfeeding was not statistically significant. Table 4 shows the classification table for both analyses. The logistic regression model classified only a small percentage of infants without breathing difficulties incorrectly, but classified only 39 of the 630 cases of breathing difficulty correctly. Since classification is sensitive to the relative number of cases, the pattern seen here is not surprising given the relative rarity of breathing difficulties (see Table 4).

### 3.3. RF Results

The RF algorithm, which included 28 variables, misclassified 2 out of the 18,552 observations in the dataset. The out of bag error rate was 0.037 with 100 iterations. The variable importance plot for the RF algorithm is displayed in Figure 1. The variable importance plot does not display causal direction, and the variable names in the figure were edited to include this information. More information on the shape of the relationship between each of the 28 variables and risk of hospitalisation is in the Appendix A (plots with Lowess trend lines). The importance plot compares the importance of all variables with the most important variable found using the algorithm, which was birthweight. Birthweight had a very substantial effect on the likelihood of breathing difficulties: for example, calculating the predictive margins from the logistic regression model, infants born at 400 g had more than a 95% likelihood of experiencing breathing difficulties in the first week after birth, while those born at 6 kg had less than a 1% likelihood. When viewed in this context, all of the variables included in the RF algorithm were somewhat important: the analysis suggested that breathing difficulties have a large number of contributing factors. Co-sleeping was not among the most important variables in the importance plot. An RF algorithm excluding co-sleeping misclassified one additional observation.

## 4. Discussion

Co-sleeping had a modest, but statistically significant main effect in the logistic regression analysis of the association with likelihood of infants experiencing breathing problems in the first week after birth. There was no evidence of a statistically significant interaction between co-sleeping and breastfeeding in the logistic regression model: there was no added advantage to both of these infant care practices together. The RF algorithm results suggested that a large number of factors underlie risk of breathing problems: no variable importance scores were close to zero; all had some predictive utility in classifying cases. Many of the variables with the highest importance scores are not easily modifiable factors: the socio-economic variables had high importance (0.5 to 0.85). The modifiable health risks of smoking and alcohol consumption, which also had high importance scores, are already discouraged in public health campaigns. Two of the RF importance score results were surprising: breast-fed infants had higher risk, as did infants whose mothers had full ante-natal care. Breast-feeding has other benefits for infants and mothers [35], and the association found here with likelihood of breathing difficulties may be explained by other factors. One such factor is that mothers who spend more time in close physical proximity with their infant may be more likely to notice that their infant is having breathing difficulties. This would make effects of variables indicative of close physical proximity underestimates of the true effects. This could also be true for co-sleeping.

Studies in public health from social and behavioural science perspectives do not often include machine learning approaches. This is not true outside of academia: in business and government machine learning has become a very frequent approach [37]. Machine learning is most useful when the number of influential variables is potentially large, and the results of the RF algorithm here indicated that there are many contributing factors in predicting infant breathing distress. On the other hand, regression modelling allowed statistical control for infants’ general health and other risk factors for poor health. This helps provide reassurance that the co-sleeping effects are specific to co-sleeping and not other correlated aspects of a child’s health. The logistic regression model had poor sensitivity: RF, which is not constrained by linear functions, produced a model with a far superior fit to the data (see Table 4).

If the proposed mechanism underlying the hypothesis is correct, that close physical proximity aids in the physiological development and maintenance of infant breathing regulation [20,21,22], bed co-sleeping will only be beneficial if it involves close physical contact. Existing safe co-sleeping guidance for parents may not promote or stimulate these physiological mechanisms. This is true of boxes distributed to new parents to separate infants in the parental bed [9]. These box distribution programs have fallen into two categories of intentions: distribution of co-sleeping boxes as part of ante-natal care incentives not specifically or solely focussed on avoiding SUDI [9], and programs targeting new mothers in groups particularly at risk for SUDI [38]. For the latter, the benefits of mother–infant separation during sleep may outweigh any advantages of close maternal–infant contact.

The present research had a large sample size that is difficult to achieve in research that had been designed specifically for studying co-sleeping. However, the disadvantages of using the MCS data were also substantial: first, mothers were not asked to provide any detail about their bed-sharing arrangements, and thus actual physical proximity of mother and infant during sleep was unknown. All that could be determined was whether bed-sharing was the most typical sleeping arrangement for the infant during the first 9 months. Second, the outcome variable was created from an interview question which did not differentiate between mild breathing difficulties and serious ones requiring hospitalization. Third, the interview was carried out nine months from birth, and due to this delay, there could be recall bias. These issues are likely to result in underestimation of the true association between co-sleeping and infant breathing distress.

## 5. Conclusions

Case-control studies identified a potential link between co-sleeping and SUDI risk [3,4]. The results reported here add to a growing number of studies suggesting that for many, co-sleeping has some advantages for infant health, stress minimisation and social functioning [9]. Co-sleeping guidance for parents needs to be balanced between avoiding SUDI risk by discouraging the least safe co-sleeping practices (such as sofa sleeping), while helping infants and mothers receive the social and health benefits that appear to be associated with co-sleeping.

## Figures and Tables

**Figure 1 ijerph-17-02985-f001:**
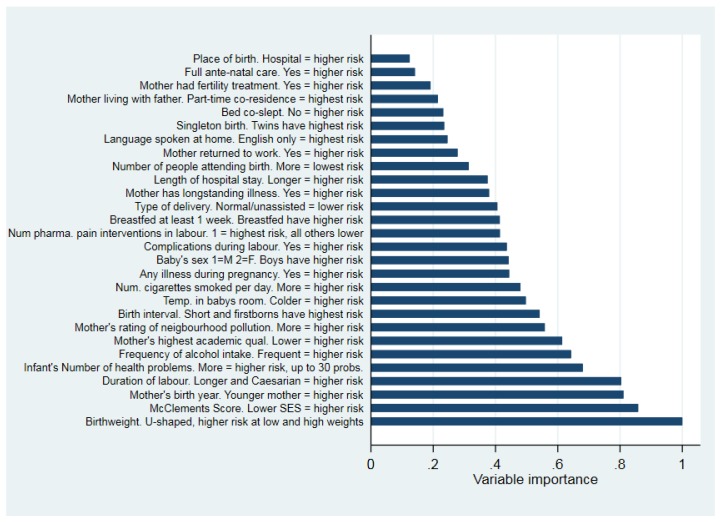
Variable importance plot from the RF model.

**Table 1 ijerph-17-02985-t001:** MCS variables included in analyses.

Variable Name	Description/Coding	Reason for Inclusion	Included in Logistic Model
Breathing distress in first week reported by parent	0 = no, 1 = yes	Dependent variable	✓
Bed co-sleeping with a parent was main sleeping arrangement in first 9 months	1 = no, 2 = yes	Main independent variable	✓
Breastfed at least 1 week	1 = no, 2 = yes	(see text)	✓
Birthweight	Recorded in kg	Predicts infant health [24]	✓
Mother’s birth year	Year of birth	Studies demonstrate risk associated with young mothers [25]	✓
Father present in household	1 = yes, 2 = part of the time, 3 = no	Associated with low maternal income and increased health risk in infants [26]	✓
Infant’s total number of illnesses reported (first 9 months)	Illnesses reported in infancy	Included to statistically control for poor general health in infancy	✓
Singleton birth, twin or triplet	1 = singleton birth, 2 = twin, 3 = triplet	Twins and triplets have poorer health outcomes in infancy [24]	✓
Household living standard	McClement’s equivalised income [27]	Socioeconomic status predicts infant health [28]	✓
Mother’s highest qualification	1 = highest, 6 = lowest. Qualifications which could not be coded assigned mid-value	Socioeconomic status predicts infant health [28]	✓
Infant’s sex	1 = Male, 2 = Female	Boys have poorer health in infancy [29]	✓
Length of hospital stay after birth	Recorded as hours, days, weeks. Categorical. 1 = shortest, 3 = longest	Mothers & infants remaining in hospital will not have the same opportunities to co-sleep as those at home.	✓
Duration of labour	Recorded in hours	Plausible predictor of neonatal health	
Mother’s alcohol consumption	Reported after birth. 1 = most frequent, to 7 = never	Predicts some infant health conditions and unsafe co-sleeping [15,16]	
Mother’s report of pollution in neighbourhood	Reported on a 4-point scale, 1 = most, to 4 = least pollution	Plausible predictor of neonatal health and breathing difficulties	
Birth interval	Categorised into quartiles. First births coded as 5 and added to scale	Often found to be associated with infant health [30]	
Mother’s smoking: number of cigarettes daily	Reported after birth	Predictor of neonatal health and breathing problems [3,31]	
Temperature in room where baby sleeps	On a 5-point scale where 1 = warmest and 5 = cold	Plausible predictor of neonatal health	
Mother returned to paid work within 9 months of birth	1 = yes, 2 = no	Plausibly interacts with co-sleeping via need for infant independence	
Complications during labour	0 = no, 1 = yes	Predictor of neonatal health [32]	
Number of pharmacological pain interventions during labour	Recorded as the number of different pain interventions	Plausible predictor of neonatal health	
Normal delivery	1 = normal, all else = 2 (including forceps, Caesarean, etc.)	Predictor of neonatal health [32]	
Maternal longstanding illness prior to pregnancy	1 = yes, 2 = no. Includes all chronic illness and disability	Predictor of neonatal health [32]	
Maternal illness during pregnancy	1 = yes, 2 = no	Predictor of neonatal health [32]	
Number of people who attended birth	Reported by mother	Measure of social support for mother and neonate	
Language other than English spoken at home	1 = English only, 2 = English plus another language, 3 = Other language only	Plausibly interacts with co-sleeping [33]	
Infant born in hospital not home	1 = yes, 2 = no	Plausible predictor of neonatal health if home births are typically low-risk	
Received full ante-natal care	1 = yes, 2 = no	Plausible predictor of neonatal health	
IVF or ART pregnancy	1 = yes, 2 = no	Predictor of neonatal health [34]	

**Table 2 ijerph-17-02985-t002:** Descriptive statistics.

	*N*	Mean	Min	Max	St.Dev
Breathing distress in first week reported by parent	18,552	0.037	0	1	0.189
Father present in household	18,525	1.115	1	3	0.447
IVF or ART pregnancy	18,545	1.974	1	2	0.159
Maternal illness during pregnancy	18,497	1.623	1	2	0.485
Infant born in hospital not home	18,502	1.021	1	2	0.142
Normal delivery	18,499	1.313	1	2	0.464
Duration of labour (hrs)	17,773	9.15	0	100	11.137
Complications during labour	18,552	0.32	0	1	0.466
Mother’s smoking	18,540	3.318	0	60	6.273
Mother’s alcohol consumption	18,529	5.133	1	7	1.49
Mother’s report of pollution in neighbourhood	18,315	3.089	1	4	0.892
Number of pharmacological pain interventions during labour	18,413	0.731	0	4	0.667
Number of people who attended birth	18,552	1.119	0	4	0.495
Infant’s sex M = 1 F = 2	18,552	1.486	1	2	0.5
Household equivalised income	16,941	296.999	14	1251	227.33
Maternal illness during pregnancy	18,524	1.789	1	2	0.408
Infant’s total number of illnesses reported	18,521	1.633	0	50	1.991
Mother returned to paid work within 9 months of birth	18,542	1.879	1	2	0.327
Mother’s highest qualification	18,484	4.005	1	6	1.304
Mother’s birth year	18,549	1971	1937	1987	5.961
Language other than English spoken at home	18,552	1.19	1	3	0.481
Singleton birth, twin or triplet	18,552	1.014	1	3	0.123
Bed co-sleeping with a parent	18,531	1.089	1	2	0.285
Birthweight	18,482	3.344	0.391	7.229	0.59
Breastfed at least 1 week	18,551	1.536	1	2	0.499
Length of hospital stay after birth	18,117	2.046	1	3	0.421
Received full ante-natal care	18,492	1.038	1	2	0.191
Temperature in room where baby sleeps	18,408	2.301	1	5	0.745
Birth interval	18,527	3.79	1	5	1.461

**Table 3 ijerph-17-02985-t003:** Logistic regression results.

Dependent Variable: Breathing Difficulty Reported In First Week Of Infancy	Odds Ratio	St.Err.	*t*-Value	*p*-Value	95% Confidence Intervals	Sig
Bed co-slept	0.694	0.115	−2.20	0.027	0.502–0.960	**
Breastfed at least 1 week	0.974	0.088	−0.30	0.767	0.815–1.162	
Coslept by breastfed interaction (centred resids.)	1.043	0.049	0.88	0.378	0.950–1.144	
Mother’s birth year	1.002	0.008	0.30	0.762	0.987–1.018	
Father present in household	0.963	0.113	−0.32	0.747	0.764–1.213	
Infant’s total number of reported illnesses	1.185	0.039	5.12	0.000	1.111–1.265	***
Infant number of illnesses squared	0.995	0.002	−2.42	0.016	0.991–0.999	**
Twin or triplet	1.290	0.265	1.24	0.214	0.863–1.928	
Income (McClement’s equivalency)	1.000	0.000	−0.91	0.365	0.999–1.000	
Mother’s educational qualifications	0.962	0.034	−1.09	0.274	0.897–1.031	
Infant sex (1 = M 2 = F)	0.695	0.060	−4.20	0.000	0.586–0.823	***
Birthweight (kg)	0.043	0.010	−13.74	0.000	0.027–0.067	***
Birthweight squared	1.504	0.058	10.65	0.000	1.395–1.621	***
Duration of hospital stay after birth (1 = longest, to 3 = shortest)	0.398	0.039	−9.50	0.000	0.329–0.481	***
Constant	126.0	83.734	7.28	0.000	34.24–463.52	***
Mean dependent var: 0.037				SD dep. var: 0.192		
Pseudo r-squared: 0.110				Num. obs: 16,491		
Chi-square: 589.501				Prob > chi2: 0.000		
Log Likelihood −2556.144						

*** *p* < 0.01, ** *p* < 0.05.

**Table 4 ijerph-17-02985-t004:** Classification results for both regression and RF models. Note that the sample sizes are different due to how missing data are handled in RF modelling.

	Observations Correctly Classified by Model	
	Logistic Regression	Random Forest
Cases with breathing difficulty	39/630 (5.87%)	687/689 (99.7%)
Cases without breathing difficulty	15,836/15,861 (99.84%)	17,863/17,863 (100%)

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
