# Peer review of "Mother–Infant Co-Sleeping and Maternally Reported Infant Breathing Distress in the UK Millennium Cohort"

_ijerph, 2020, doi:10.3390/ijerph17092985_

Round 1
Reviewer 1 Report
This study uses an extensive database from the UK Millenium Cohort Study to answer a novel question, which would not usually be able to be addressed without assistance of a large cohort.
I believe these are interesting findings and certainly consistent with previous evidence that supports that shared sleeping between infants and their caregivers is the biological norm, that is still common practice throughout the world, and in fact practised by 75% of Qld families (see Colet et al 2019) with around 50% regular shared sleepers. These recent articles from an Australian cohort may strengthen this background argument/rationale.
- COLE R, YOUNG J, KEARNEY L, THOMPSON JMD. (2020) Reducing sleep-related infant mortality through understanding factors associated with breastfeeding duration: a cross-sectional survey. Breastfeeding Review (Submitted 24 June 2019, Accepted Mary Paton Prize Best Paper 2019, $5000, accepted March 2020)
- COLE R, YOUNG J, KEARNEY L, THOMPSON JMD. (2020) Infant care practices and parent uptake of safe sleep messages: a cross sectional survey in Queensland, Australia. BMC Pediatrics 20: 27. https://doi.org/10.1186/s12887-020-1917-5.
However several integral modifications are required for this manuscript to make a meaningful contribution to the literature.
The background information is somewhat relevant however there are several key corrections to address in order not to perpetuate misinformation for health professionals and the public.
- Please use appropriate terminology and nomenclature in referring to SIDS and SUDI throughout the manuscript
Firstly, sudden infant death does not equate to only SIDS - please ensure that all references to SIDS are replaced with the more appropriate term Sudden Unexpected Death in Infancy (SUDI) which includes sudden infant death syndrome (SIDS) and fatal sleeping accidents (W65); inclusive of all R95-R99 codes and W65 on the ICD-10 Causes of Death classification. Referring to SIDS when in fact many of these deaths are attributed not to SIDS by definition but to Undetermined or SUDI, further undermines the effectiveness of safe sleep public health campaign messages and perpetuates myths with both parents and health professional consumers of this information. Please define and use the terms appropriately; these are not interchangeable; SIDS is a subset of SUDI.
- The author purports that health organisation and health systems publish guidelines that suggest that babies should sleep in a cot rather than a bed with their parents, and references American guidelines. However this does not accurately represent the current status of guidelines in many countries outside of the USA which support risk minimisation.
The reference used for the NHS refers to only one NHS trust and does not link to the guideline. The UK NHS supports the Lullaby Trust guideline which provides a risk minimisation approach, as does Red Nose Australia. Risk minimisation approach acknowledges that some parents choose to share a bed or sleep surface or may not intend to, but do fall asleep.
The relevance of the manuscript background to why this study’s results are important to share could be greatly strengthened by acknowledging the wealth of work that has been done in the last 10-15 years in Australia, UK and NZ that supports a risk minimisation approach to mother-infant shared sleeping in terms of strategies to reduce risk in shared environments, and professional guidelines that support risk minimisation. Please see the Australian College of Midwives, the Australian Breastfeeding Association, the Lullaby Trust, the American Academy of Breastfeeding Medicine; all provide recommendations for risk minimisation rather than actively recommending against shared sleeping in all situations.
https://www.lullabytrust.org.uk/safer-sleep-advice/co-sleeping/
https://www.midwives.org.au/resources/co-sleeping-and-bed-sharing-acm-position-statement-2014
https://www.midwives.org.au/sites/default/files/uploaded-content/field_f_content_file/acm_cosleeping_lit_review_20141216_0.pdf
The introduction could acknowledge the debate that still exists in terms of risk elimination (USA approach) versus risk elimination (Australia, UK and NZ) rather than stating erroneously that (all) health organisations and health systems publish guidelines that recommend against bedsharing - this is a misrepresentation. Please see Young and Shipstone, 2019.
- YOUNG J, SHIPSTONE R. (2018) Shared Sleeping Surfaces and Dangerous Sleeping Environments [Chapter 11]. In BYARD RW, DUNCAN J. (Eds.) SIDS, Sudden Infant and Early Childhood Death: The Past, The Present and the Future. 1st Adelaide: University of Adelaide Press. Available at https://www.adelaide.edu.au/press/titles/sids/sids-ebook.pdf
- Hypothesis - The hypothesis presented (page 2 of 11) line 71-74, stated that ‘that cosleeping infants will be less likely to be admitted to hospital with breathing difficulties.’
The hypothesis should be reframed.
The outcome variable of interest is infant breathing difficulties up to a week from birth, and appears to be based on maternal recall of her infant care practices when the infant is 9 months of age (page 2 of 11, line 82).
There is no measure of hospital readmission post birth that I could find in this paper and therefore the proposed hypothesis seems irrelevant and was not addressed in the discussion or implications for this study moving forward - this hypothesis should be reframed in terms of what was measured.
Please define the time period for which parents were reporting that ‘cosleeping with a parent as normal practice’ - this was not clear. Was this any time in the first 9 months of life (when first questionnaire appears to have been administered) -in which case please provide this detail, and comment in limitations on memory and recall bias for parents reporting on infant care practices in the first week of life at 9 months post birth. OR is this meant to represent cosleeping practices in the first week of life.
There was no anatomically or physiologically plausible explanation presented in the methodology for why an experience of breathing difficulties in the first week of life would be in any way influenced by what occurred when the infant was discharged home and shared a sleep space with their parent in bed as their normal sleeping arrangement, if this was for their usual shared sleeping during the first 9 months of life - this is not clear. It was also stated that main sleeping arrangement was captured rather than atypical environments (eg when infant was sick and may have respiratory difficulties). While such an examination would make more sense if kangaroo care/skin to skin contact was being examined for its association on breathing difficulties particularly in the first week of life for sick and premature infants, this particular examination of breathing difficulties experienced in first week of life as measured by parental recall, with influence of usual sleeping environment for infant over a longer period and after discharge post birth, is tenuous as best and appears only to be a data mining exercise.
- Defining variables
Infant breathing difficulties needs to be defined. What constituted maternal reports of infant breathing difficulty. This could vary greatly depending on previous maternal experience, parity, previous infant illness etc; from requiring ventilation (to survive) to being a bit snuffly when feeding(normal).
Infant Gestational age, which would be strongly associated with experience of infant breathing difficulties during the first week of life, was not discussed or included as a variable. Was this information available. That the minimum birth weight was 391 grams (extremely premature and/or growth restricted) and maximum was 7229 grams (macrosomia, maternal diabetes) suggests that babies on the edge of viability were included in this cohort; these babies would have almost certainly required neonatal intensive care and respiratory support for prolonged periods. Maternal recall of breathing difficulties during first week of life, a definite or likely yes in these cases, would not be influenced or associated with subsequent practices of shared sleeping beyond discharge from hospital, if and when these babies were eventually discharged from hospital.
Limitations of this study need to be discussed transparently: maternal recall bias should be discussed; limitations of preset collected variables etc.
- Comments about use of boxes distributed to new parents needs to be differentiated in order to ensure evaluated programs are not represented. Please differentiate between a)commercial initiatives that only involve distribution of a cardboard box not designed for use in the parental bed (Finnish Baby Box derivatives, which have not yet been evaluated in terms of safety or purpose) and the b) Pepi-Pod and Wahakura Programs which provide a sleep space, safe sleep education and a family commitment to using the Pod as intended and which have been designed to be used IN the parental bed to maintain maternal infant closeness and contact while providing a zone of protection for baby’s airway - eg Qld Pepi-Pod Program, NZ Pepi-Pod and Wahakura Programs have been evaluated as safe, feasible and culturally appropriate programs to support traditional shared sleeping practices.
An explicit ethical review statement was not included; simply stated that data from the MCS data are freely available from the UK Data Service.
Author Response
Reviewer 1’s concerns centred on five points. First, the terminology for sudden infant death in the introduction did not encompass all unexplained infant death. I have changed this by replacing SIDS with SUDI throughout the manuscript (lines 32-39 and elsewhere in the manuscript document with Tracked Changes showing).
Second, there was concern around how the discussion of published co-sleeping guidelines omitted risk-minimisation approaches: while the NHS and CDC both have current guidelines discouraging mother-infant co-sleeping, a number of other organisations accept co-sleeping as a normal, healthy behaviour but publish guidelines on how to safely co-sleep. The first part of the introduction (lines 29-49) has been edited so that the statements about published guidelines include risk-minimisation. References for these have also been added.
Third, the last paragraph of the introduction contained an incorrect framing of the hypothesis. This has been edited (lines 82-83). The second part of the comment by Reviewer 1 and their fourth concern was how co-sleeping and breathing difficulties were specifically defined in the research. More detail is provided on lines 106-110, and the shortcomings of MCS survey methodology with regard to these variables is discussed in lines 238-244. I additionally edited the manuscript title so that it is clear that the data are from maternal reports.
The fifth comment from reviewer 1 was on recall bias, which is now discussed on lines 244-247.
Reviewer 2 Report
This is a potentially important and topical study exploring the idea that mother-infant co-sleeping, generally not recommended due to SIDS could positively impact other aspects of child health - in this case infant breathing distress.
I have one major problem that i feel needs to be more strongly addressed;
How can such low quality evidence for breathing distress bue used - issues include:-
a dichotomous variable
determined by an untrained person's subjective opinion
for the bulk of the time the perons would be asleep in the "test period"
of a short (1 week) sample
all from recall of events 9 months in the past
in a group with very limited numbers of cases of reported distress
I think an argument needs to be made as to how such problematic evidence can still remain useful to draw any conclusions.- the effect is tiny, variable, heavily confounded and seems only significant due to the huge numbers involved in the calculations.
Further, the mechanisms are tenuous to say the least - more a just so story
and i feel breathing distress needs to be more clearly defined.
I actually agree with the authors aguments and conclusions but feel strongly that they need to be strengthened by better arguments and evidence
good luck
Author Response
Reviewer 2 was concerned primarily about data quality. The discussion now includes this point: by using a general survey of infant health rather than a research design targeted at co-sleeping, data quality was partly traded-off for data quantity. On lines 246-247 I state the likely statistical effect of this trade-off, which is underestimation of the true association between co-sleeping and infant breathing difficulties.
Both reviewers felt that the physiological mechanisms were tenuous, although the second to last paragraph of the introduction describes research specifically linking the physiological regulation of breathing in early infancy with regard to maternal physical contact. The reference tying this literature together is provided (Trevathan and McKenna, 1994). I have made a clearer statement that the research evidence on maternal physical contact (at night) implies that infants who do not co-sleep with close maternal contact will not have this maternal influence on breathing regulation during sleep (lines 80-85). This is an important point: it is why the focus was on early infancy, and on breathing difficulties rather than other aspects of infant health early in life.
Round 2
Reviewer 1 Report
Most Key areas highlighted in the original review have been addressed appropriately. Appropriate references have been added to the background, further clarification around questions used from the Millenium Cohort have been provided and limitations of using this data, not designed to answer specific questions about shared sleeping arrangements have been discussed.
Two key issues require further discussion:
Page 9 of 12: new reference provided relating to NHS Essex Trust Bedsharing guidelines is in contrast to the CDC guideline in the same sentence. The difficulty is that each NHS trust, like Australian States and Territories have different guidelines - some are risk minimisation (that support shared sleeping with risk reduction measures in place recognising this is likely to occur once home and commencing education in hospital) and others are risk elimination (just stated do never bedshare or cosleep).
Clarification is required around reference to 'boxes' distributed to new parents. Baby Box programs (which do not differentiate between families with vulnerabilities and those with low risk) are different in construction and purpose to wahakura and Pepi-Pod Programs which are specifically targeted to families with increased risk of SUDI including increased risk of fatal sleeping accidents. These programs have demonstrated cultural acceptance as similiar to traditional practices. eg premature infants of low birth weight with caregivers who smoke are at a much higher risk of SUDI and benefit from close proximity however within a zone of physical protection to protect their vulnerable airway.
If mentioning 'box' programs, rather than contribute to confusing messages in the literature about something that is not within the field of expertise or directly relating to this study at least differentiate between 'baby box' programs (originated in Finland as part of an incentive to attend antenatal care) and trademarked Pepi-Pod and wahakura programs - these have different purposes and are part of a program that specifically provides education, support and family engagment in safe sleep practices, not just a sleep space.
2) Please clarify the period in which cosleeping or shared sleeping data is collected which is used to explore the association with maternal reports of breathing distress in the first week of life. Please clarify Page 3 of 12: Line 11-113:
The main predictor variable was whether or not the infant co-slept with a parent in bed as their normal sleeping arrangement, during xxxxx? (also during the first week of life, up until the 9 month interview? Please explicity state this time period to give this variable context). Please also clarify this in Table 1 - currently only stated this as Bed cosleeping with a parent ? When, in first week of life; also infant illnesses reported - are these in first week of life or during the first 9 months of life? Please clarify in Table 1.
If this information is not available from this Cohort Dataset this needs to be discussed as a considerable limitation of the study.
Author Response
Reviewer 1 made two additional suggestions. 1. To clarify the description of the use of baby boxes on page 9 of the manuscript. I have added detail about baby box programs as either directed at reducing SUDI specifically, or as programs that are not directed at reducing SUDI. In addition I have added a statement to the introduction to highlight that not all NHS Trusts discourage co-sleeping. 2. The co-sleeping variable was not adequately defined. I have addressed this in the variable description table and in the discussion (lines 246-247).
Reviewer 2 Report
all concerns addressed adequately, good work
Author Response
Reviewer 2 had no further comments.